# The Influence of Camera and Optical System Parameters on the Uncertainty of Object Location Measurement in Vision Systems

**DOI:** 10.3390/s20185433

**Published:** 2020-09-22

**Authors:** Jacek Skibicki, Anna Golijanek-Jędrzejczyk, Ariel Dzwonkowski

**Affiliations:** Faculty of Electrical and Control Engineering, Gdańsk University of Technology, Narutowicza 11/12, 80-233 Gdańsk, Poland; anna.golijanek-jedrzejczyk@pg.edu.pl (A.G.-J.); ariel.dzwonkowski@pg.edu.pl (A.D.)

**Keywords:** vision measurement systems, measurement uncertainty, location measurement systems

## Abstract

The article presents the influence of the camera and its optical system on the uncertainty of object position measurement in vision systems. The aim of the article is to present the methodology for estimating the combined standard uncertainty of measuring the object position with a vision camera treated as a measuring device. The identification of factors affecting the location measurement uncertainty and the determination of their share in the combined standard uncertainty will allow determining the parameters of the camera operation, so that the expanded uncertainty is as small as possible in the given measurement conditions. The analysis of the uncertainty estimation presented in the article was performed with the assumption that there is no influence of any external factors (e.g., temperature, humidity, or vibrations).

## 1. Introduction

Vision systems have been very popular for over 20 years. They are used not only in military solutions (biometric systems, automatic missile guidance, reconnaissance systems) and technology [1,2] (among others, modern human–computer interfaces, examination of object features, sorting products, inspection of dimensions and contours, checking correctness and completeness of product performance, food control [3]), but also in medicine (laboratory tests [4], surgical procedures, telemedicine), cartography and ecology (site map analysis for mineral exploration or pollution monitoring), transport (rail [5,6,7,8,9,10], air), the exploration of the Earth and the Universe (interpretation of satellite and astronomical images), and security measures and surveillance, such as reading license plates, detecting explosives at airports, and monitoring crowd behaviour.

As it can be seen, due to their universality, these systems can be found in virtually every field of technology. The advantages of this technology include its increasing reliability, its increasing ease of use, its non-contact measurement method with high accuracy, and the decreasing cost of systems, the latter resulting in measurable economic benefits. A vision system is a set of cooperating electronic devices that is designed to enable visual inspection and analysis of an object or environment, similar to human eyesight.

Video and location measurements are a particular application of vision technology. The results of these measurements are used, among others, for detecting the position of a workpiece machined on a Computerised Numerical Control (CNC) machine or the displacement of contact wires in railway transport [11,12].

The measurement result should be presented together with the quality parameter of this measurement, which is either a measurement error or measurement uncertainty. The measurement uncertainty is a parameter related to the measurement result, defining the range of measured quantity in which the actual value of the measured quantity is located with a certain probability [13]. Only then do the results provide information about the quality of the measurements performed, and only then can they be compared with each other. The results presented without a qualitative measure are incomplete from the metrological point of view and therefore are useless. This results from both the requirements that have been valid for more than 20 years [13,14,15,16] and also from the fact that the measurement uncertainty has an advantage over the measurement error. The advantage is that more information about the measurement (distribution of measurement results), as well as about the value of the coverage factor and the probability of expansion is obtained, e.g., 95%. For this reason, regardless of whether the measured quantity is electric [17] (e.g., current [18,19] or power measurement [20,21,22]) or non-electric (e.g., temperature [23,24,25], pressure [26], mass flow [27,28,29], and time [7]), an estimate of the uncertainty of the measured value is provided.

The uncertainty determines the range surrounding the measurement result containing a large, predetermined portion of the results that can be attributed to the measured quantity. This interval is called the range of uncertainty of the measurement result.

It should be remembered that if there are systematic disturbances in the experiment, then according to good metrological practices and Guide to the Expression of Uncertainty in Measurement (GUM) recommendations [13,14,15,16], the observation results should first be corrected by introducing corrections to mitigate the influence of these disturbances, and then, uncertainty estimation analysis should be performed.

There are many works on vision systems or camera systems that assess the measurement uncertainty: mechanics, machine-vision [30,31,32], flow measurements [33,34,35], geoinformation sciences [36,37,38], or medicine [39]. Naturally, in each of the presented examples, the authors performed estimates of quality parameters, most often as measurement errors or standard deviations. At the same time, the impact of one particular parameter on the measurement accuracy was the most frequently studied factor.

You can also find publications on the assessment of uncertainty in measuring the location of an object, where the source is a camera treated as a measuring device [9,11,40]. Existing publications only cover a general discussion on this issue.

In the presented publication, the authors made a detailed analysis (based on experimental research) of the impact of several parameters, such as the brightness level of the recorded images, the sensitivity of the image sensor, and the focal length of the lens on the accuracy of object estimation on the image sensor.

Moreover, the article shows how to determine the complex uncertainty in measurement of the object’s location, which makes this article stand out from among the previous ones.

Naturally, the image measurement uncertainty in the camera is affected by image calibration and image dewarping. Numerous methods are currently used for image calibration, including performing object position calibration and colour calibration [41], with the use of the Total Least Square (TLS) and Feedforward Neural Network algorithm [42], and double-DAC (Digital to Analog Converter) interlaced image calibration (TIDAC) with the use of machine learning [43].

Moreover, in the case of dewarping, in addition to traditional algorithms using the extraction and segmentation of object features, several modified methods of removing image distortion are used, e.g., from thick to thorough drainage with the use of deep learning methods [44,45,46], a reliable estimation of curled lines of text [47], as well as employing Scale-Invariant Feature Transform (SIFT) transformation [48] or stochastic calculations in combination with the neuromorphic system [49].

The measurement method proposed by the authors does not require image calibration. The measurement result together with the uncertainty is obtained based only on the analysis of the recorded image and the characteristic dimensions of the measuring stand.

The identification of factors that cause the uncertainty of the position measurement and the determination of the degree of their contribution to the combined standard uncertainty of position measurement will allow for the camera parameters to be selected in such a way that the expanded uncertainty is as small as possible in the given measurement conditions.

The camera is not a typical measurement device, for which the producer determines the accuracy level and presents it in technical documentation. For this reason, it is necessary to estimate the measurement uncertainty in a different way. When measuring the position of an object in the video space with the measuring system, the camera uncertainty is understood as the uncertainty of measuring the object position on the image sensor. The purpose of this article is to present the methodology for estimating this uncertainty and provide a full metrological analysis, as well as determine the impact of selected parameters on the value of this uncertainty. This procedure is universal and can be used for any type of camera.

Section 2 of this article describes the measurement conditions when a video camera is used, and the assumptions of the experiment, whose results are analysed in the following chapters. In Section 3, the derivation of a theoretical equation for combined standard uncertainty in the measurement of an object position on the image sensor in *x* and *y* axes was presented. Then, the factors influencing the measurement uncertainty, such as the brightness level of the recorded images, sensitivity of the image sensor, and focal length of the lens are presented (Section 4). The fifth Section shows the influence of selected, considered factors on the real measurements results, based on the example of measurements of vertical displacements of the high-voltage overhead power line conductor. Some final conclusions are provided in Section 6.

Some of the results contained in this paper were partially presented in the book [50] written in the Polish language.

## 2. Principle of Measuring

The article analyses the factors affecting the uncertainty of the image position on the image sensor, using the stand whose diagram is shown in Figure 1.

For the discussed experiment, the aim of the analysis is to determine the uncertainty of the measurement of the object image position on the image sensor, as shown in Figure 2.

When measuring displacements in an optical way with the use of a camera, regardless of the position configuration and the related dependence on the measurement result, one of its components is always the position of the object image on the image sensor.

The following assumptions were made:the same factors influence the accuracy of the point displacement on the image sensor in both axes,constant mapping scale,the camera is not affected by any external factors such as change in temperature, humidity, or vibrations.

For the considered configuration of the measurement stand, the value of the position of the object measured in the horizontal axis *OX*, determined based on the position of its image on the image sensor, is given by the following dependence:(1)$$x=x^{\prime }\cdot \left(\frac{k}{F}-1\right).$$

Due to the parallelism of the object plane and the image plane, the analogous relationship is valid for displacements in the vertical axis *OY*:(2)$$y=y^{\prime }\cdot \left(\frac{k}{F}-1\right).$$

The distance *F* between the optical centre of the lens and the image plane depends on the focal length of the lens and the scale of reproduction, and it is given by the dependence [11]:(3)$$F=\frac{k-\sqrt{k^{2}-4\cdot k\cdot f}}{2}.$$

The focal length of the lens *f*, even with fixed focal length lenses, is not a constant value but varies slightly depending on the current focus setting. There is a so-called focal length floating. This effect must be taken into account. Therefore, the focal length of the lens is determined indirectly based on an additional measurement for the current lens focus in accordance with the formula [11]:(4)$$f=\frac{k}{2+\frac{{x^{\prime }}_{w}}{x_{w}}+\frac{x_{w}}{{x^{\prime }}_{w}}}.$$

## 3. Uncertainty of Measuring Objects with Imaging Camera

The uncertainty of the position measurement in the *x*-axis, according to the law of uncertainty propagation [13,15,16], is defined by the formula (for the *y*-axis the uncertainty analysis will be the same):(5)$$u\left(x\right)=\sqrt{{\left(\frac{\partial x}{\partial x^{\prime }}\right)}^{2}\cdot u{\left(x^{\prime }\right)}^{2}+{\left(\frac{\partial x}{\partial k}\right)}^{2}\cdot u{\left(k\right)}^{2}+{\left(\frac{\partial x}{\partial F}\right)}^{2}\cdot u{\left(F\right)}^{2}}.$$

The sensitivity coefficients in Formula (5) for the *x*-axis are respectively:(6)$$\frac{\partial x}{\partial x^{\prime }}=\frac{k}{F}-1$$
(7)$$\frac{\partial x}{\partial k}=\frac{x^{\prime }}{F}$$
(8)$$\frac{\partial x}{\partial F}=\frac{-x^{\prime }\cdot k}{F^{2}}.$$

By introducing the above-described sensitivity coefficients into Formula (5), the following dependence was obtained, defining the standard uncertainty of the object position measurement for the *x*-axis:(9)$$u\left(x\right)=\sqrt{{\left(\frac{k}{F}-1\right)}^{2}\cdot u{\left(x^{\prime }\right)}^{2}+{\left(\frac{x^{\prime }}{F}\right)}^{2}\cdot u{\left(k\right)}^{2}+{\left(\frac{-x^{\prime }\cdot k}{F^{2}}\right)}^{2}\cdot u{\left(F\right)}^{2}}.$$

It is known that the distance *F* between the optical centre of the lens and the image plane depends on the focal length and the scale of the projection, and it is given by (3), where the focal length *f* describes Formula (4) in which *x*′_w_ is the image size of the reference object with known *x*_w_ dimensions on the image sensor. The object is located at the distance *F* from the image sensor.

By introducing the formula for *f* (4) into the dependence (3), after the transformations, the following dependence was obtained:(10)$$F=\frac{k}{2}\cdot \left(1-\sqrt{1-\frac{4\cdot x_{w}\cdot {x^{\prime }}_{w}}{{\left(x_{w}+{x^{\prime }}_{w}\right)}^{2}}}\right).$$

This formula links the distance *F* between the optical centre of the lens and the image plane to the image size of the reference object *x*′_w_. In accordance with the law of uncertainty propagation, the uncertainty of the *F* determination was defined as follows (assuming there is no correlation between the uncertainties of measured values):(11)$$u\left(F\right)=\sqrt{{\left(\frac{\partial F}{\partial k}\right)}^{2}\cdot u{\left(k\right)}^{2}+{\left(\frac{\partial F}{\partial x_{w}}\right)}^{2}\cdot u{\left(x_{w}\right)}^{2}+{\left(\frac{\partial F}{\partial {x^{\prime }}_{w}}\right)}^{2}\cdot u{\left({x^{\prime }}_{w}\right)}^{2}}$$
where the sensitivity coefficients are given, respectively, by the formulas:(12)$$\frac{\partial F}{\partial k}=\frac{1}{2}\cdot \left(1-\sqrt{1-\frac{4\cdot x_{w}\cdot {x^{\prime }}_{w}}{{\left(x_{w}+{x^{\prime }}_{w}\right)}^{2}}}\right)$$
(13)$$\frac{\partial F}{\partial x_{w}}=\frac{k\cdot {x^{\prime }}_{w}}{x_{w}^{2}-{x^{\prime }}_{w}^{2}}\cdot \left(1-\frac{2\cdot x_{w}}{x_{w}+{x^{\prime }}_{w}}\right)$$
(14)$$\frac{\partial F}{\partial {x^{\prime }}_{w}}=\frac{k\cdot x_{w}}{x_{w}^{2}-{x^{\prime }}_{w}^{2}}\cdot \left(1-\frac{2\cdot {x^{\prime }}_{w}}{x_{w}+{x^{\prime }}_{w}}\right).$$

After introducing Relations (12)–(14) into Formula (11) concerning the uncertainty of the distance measurement *F*, it will take the form:(15)$$u\left(F\right)=\sqrt{\begin{array}{l} {\left[\frac{1}{2}\cdot \left(1-\sqrt{1-\frac{4\cdot x_{w}\cdot {x^{\prime }}_{w}}{{\left(x_{w}+{x^{\prime }}_{w}\right)}^{2}}}\right)\right]}^{2}\cdot u{\left(k\right)}^{2}+{\left[\frac{k\cdot {x^{\prime }}_{w}}{x_{w}^{2}-{x^{\prime }}_{w}^{2}}\cdot \left(1-\frac{2\cdot x_{w}}{x_{w}+{x^{\prime }}_{w}}\right)\right]}^{2}\cdot u{\left(x_{w}\right)}^{2}+\\ +{\left[\frac{k\cdot x_{w}}{x_{w}^{2}-{x^{\prime }}_{w}^{2}}\cdot \left(1-\frac{2\cdot {x^{\prime }}_{w}}{x_{w}+{x^{\prime }}_{w}}\right)\right]}^{2}\cdot u{\left({x^{\prime }}_{w}\right)}^{2} \end{array}}.$$

Standard uncertainties in measuring the sizes *k* and *x*_w_ result from the accuracy of measuring instruments and are described by the following formulas (assuming a uniform probability distribution):(16)$$u\left(k\right)=\frac{\Delta k}{\sqrt{3}}$$
(17)$$u\left(x_{w}\right)=\frac{\Delta x_{w}}{\sqrt{3}}.$$

The standard uncertainty of the *x*′_w_ measurement results from the possibility of determining the image dimension of the reference object on the image sensor. This size is the dimension of the object on the recorded image expressed in pixels, and one of the main sources of uncertainty will be the inaccuracy of reading the result by the experimenter.

First, we assume that the image size of the reference object on the image sensor *x*′_w_ can be determined in accordance with the following formula:(18)$${x^{\prime }}_{w}=n_{\mathrm{pix}}\cdot l_{\mathrm{pix}}.$$

We also assume that the experimenter, determining the image size of the object based on the assessment of the photographic frame registered by the camera, may make an error equal to one pixel, due to the perceptual abilities of human sight when assigning the boundary edge of the image of the reference object to a particular pixel. With such assumptions, the following formula can be written:(19)$$u\left({x^{\prime }}_{w}\right)=\frac{\Delta {x^{\prime }}_{\mathrm{we}}}{\sqrt{3}}.$$

After introducing Dependencies (16)–(19) into Formula (15) on the uncertainty of the object position measurement in the horizontal axis *x*, it will look as follows:(20)$$u\left(x\right)=\sqrt{\begin{array}{l} {\left(\frac{k}{F}-1\right)}^{2}\cdot u{\left(x^{\prime }\right)}^{2}+{\left(\frac{x^{\prime }}{F}\right)}^{2}\cdot {\left(\frac{\Delta k}{\sqrt{3}}\right)}^{2}+{\left(\frac{-x^{\prime }\cdot k}{F^{2}}\right)}^{2}\cdot \\ \cdot \left[\begin{array}{l} {\left[\frac{1}{2}\cdot \left(1-\sqrt{1-\frac{4\cdot x_{w}\cdot {x^{\prime }}_{w}}{{\left(x_{w}+{x^{\prime }}_{w}\right)}^{2}}}\right)\right]}^{2}\cdot {\left(\frac{\Delta k}{\sqrt{3}}\right)}^{2}+{\left[\frac{k\cdot {x^{\prime }}_{w}}{x_{w}^{2}-{x^{\prime }}_{w}^{2}}\cdot \left(1-\frac{2\cdot x_{w}}{x_{w}+{x^{\prime }}_{w}}\right)\right]}^{2}\cdot {\left(\frac{\Delta x_{w}}{\sqrt{3}}\right)}^{2}+\\ +{\left[\frac{k\cdot x_{w}}{x_{w}^{2}-{x^{\prime }}_{w}^{2}}\cdot \left(1-\frac{2\cdot {x^{\prime }}_{w}}{x_{w}+{x^{\prime }}_{w}}\right)\right]}^{2}\cdot {\left(\frac{\Delta {x^{\prime }}_{\mathrm{we}}}{\sqrt{3}}\right)}^{2} \end{array}\right] \end{array}}.$$

The formula in the *y*-axis will be analogous. In Dependence (20) and the analogous formula for the *y*-axis, all their components can be determined except for *u*(*x*′) and *u*(*y*′), i.e., the uncertainty of the standard measurement of the object position on the image sensor in the *x* and *y* axes, the designation of which is the purpose of this article.

## 4. Factors Determining Uncertainty of Object Location Measurement

Factors affecting the uncertainty values in the standard measurement of the object position on the image sensor *u*(*x*′) and *u*(*y*′) in the *x* and *y* axes are:registration parameters, translating into the brightness level of recorded images,features of the camera and the optical system, such as the sensitivity of the image sensor and the focal length of the lens.

Due to the fact that the measurement experiment did not show any correlation between the above uncertainties, they can be treated as mutually independent. Therefore, the combined standard uncertainty *u*(*x*′) is described by the dependence [13]:(21)$$u\left(x^{\prime }\right)=\sqrt{u_{\mathrm{reg}}^{2}\left(x^{\prime }\right)+u_{\mathrm{cam}}^{2}\left(x^{\prime }\right)}.$$

To estimate the uncertainty associated with the characteristics of the camera and the *u*_cam_(*x*′) optical system, one has to determine the effect of the current sensor sensitivity on the uncertainty in measuring the image position of the *u*_sns_ object (*x*′) and the effect of focal length change on the uncertainty while maintaining the *u*_f_(*x*′) scale, in accordance with the formula:(22)$$u_{\mathrm{cam}}\left(x^{\prime }\right)=\sqrt{u_{\mathrm{sns}}^{2}\left(x^{\prime }\right)+u_{f}^{2}\left(x^{\prime }\right)}.$$

The final dependence on the uncertainty *u*(*x*′) in measuring the position of the object image on the image sensor in the horizontal axis *x* will take the form:(23)$$u\left(x^{\prime }\right)=\sqrt{u_{\mathrm{reg}}^{2}\left(x^{\prime }\right)+u_{\mathrm{sns}}^{2}\left(x^{\prime }\right)+u_{f}^{2}\left(x^{\prime }\right)}.$$

The uncertainty for the vertical axis *y* is determined analogously.

To verify the theoretical assumptions, a series of measurements showing the impact of particular factors on the uncertainty of the object position on the image sensor was performed. All the measurements were carried out for a fixed reproduction scale, for which the measuring range was ±12 cm in both axes, and the object measured was a 10 mm diameter round-shaped spot contrasting with the background. In total, for verification, 20 measurement series were performed, each of which contained 10,000 individual measurements. The tests were performed using a Basler 2D image camera, type acA 2040–180 kc, with the following basic technical data: sensor resolution: 2046 × 2046 px (4 Mpix); sensor size: 11.26 × 11.26 mm; dimension of a single pixel 5.5 × 5.5 μm; maximum registration speed: 180 fps.

In addition to the camera, several lenses were used. The focal length of the lenses was selected according to need. In order to reduce the impact of lens distortion on the measurement results, only lenses with a high degree of correction were used, so that their distortions, especially the barrel and pincushion distortions, were negligible and did not have a significant impact on the measurement results. This is how the need to calibrate the camera, which is necessary in the presence of lens distortions, was avoided [51,52]. A colour image in RGB standard was recorded. Then, a red channel was extracted from it (due to the low colour temperature of the light source-incandescent light), resulting in a black and white image in grayscale. Next, the colour threshold procedure was performed, as a result of which a binary image was obtained. Then, the image was subjected to morphological transformations (erosion and dilatation), as well as to partial filtration, in order to remove small artefacts and objects whose position was not measured. The centre of mass was taken as the position of the object image on the image sensor. All the above-mentioned operations, i.e., recording, processing and image analysis, were carried out using the LabVIEW software [53].

The view of the measuring stand is presented in Figure 3.

### 4.1. Uncertainties of the Camera Measurement Related to the u_reg_ Image Registration Parameters (x′)

Uncertainty studies related to the image recording parameters *u*_reg_ (*x*′) and the sensor parameters *u*_sns_(*x*′) resulting from the intrinsic sensitivity of the sensor were carried out for the distance *k* = 1543.06 ± 0.87 mm, which required the use of the Helios 44-2 2/58 lens with the focal length *f* = 54.41 ± 0.04 mm. The distance *F* was 60.827 ± 0.044 mm. The quantities *k*, *f*, and *F* were measured and determined according to Formulas (3) and (4), using a Bosh GLM 80 laser rangefinder and FWP MADb 400 calliper. Measurement uncertainties for these quantities were determined in accordance with the principles presented in the standard [13].

The adjustment of recorded image parameters can be realised, as in the case of any image acquisition by using a video or photo camera, by changing three parameters: the exposure time, lens diaphragm, or sensor sensitivity. The exposure time of a single frame is conditioned by the dynamics of registered position changes, which is a superior parameter, determining the camera speed. Hence, it is not possible to adjust the image acquisition parameters by lengthening or shortening the exposure time, or at best, this possibility is very limited. Adjustment by changes of the lens aperture value does not affect the camera work and, therefore, it does not affect the measurement accuracy. However, such an effect occurs when the parameters of the acquisition image are adjusted by changing the sensor sensitivity.

In order to estimate the effect of image recording parameters on the measurement uncertainty of the camera, registrations were made for various settings of the exposure time of the frame and the aperture value of the lens, while maintaining a constant value of sensor sensitivity. These changes translated into different levels of brightness of the recorded image. The registration was carried out starting from the settings characteristic for the optimal level of image brightness, toward its increase and decrease. Brightness level changes were determined in relative EV units in relation to the optimal level. The EV scale is logarithmic; i.e., each increase in the brightness level by +1 EV means that twice as much light as before the change was applied to the camera sensor. Naturally, changing the EV value can be achieved either by adjusting the exposure time and aperture value of lens or by changing both these parameters simultaneously. The obtained measurement results are presented in Figure 4a. To improve the readability of the drawing, the measurements made for individual lighting levels are shown in different colours.

Figure 4b presents the uncertainty *u*_reg_(*x*′) for the measurement in the horizontal and vertical axis as a function of the brightness change of the recorded image, which is calculated as a standard deviation in accordance with the formula:(24)$$u_{\mathrm{reg}}\left(x^{\prime }\right)=\sqrt{\frac{1}{\left(n-1\right)}\cdot {\sum_{i=1}^{n}\left(x_{i}-\bar{x}\right)}^{2}}.$$

It can be observed that the obtained measurement results significantly depend on the change in the brightness of the recorded image, resulting from the change of registration parameters. Changing the brightness level within the range from −1 to +2 EV in relation to the level of optimal brightness increases the uncertainty *u*_reg_(*x*′) to the level that does not exceed twice the value obtained in optimal conditions. Such deterioration in the quality of the obtained results can be considered acceptable in technical measurements. A further change in the level of brightness causes an avalanche increase in the standard uncertainty *u*_reg_(*x*′), which is particularly visible in the case of its increase, where the value of +2.2 EV is the limit level, above which the correct interpretation of the recorded image is impossible. A change in the recording parameters in the direction of decreasing brightness causes a faster increase in the uncertainty *u*_reg_(*x*′), but with a smaller gradient of changes. For the level of −2 EV, its value is almost seven times higher than at the optimum level, and a change in brightness below −2.5 EV results in a such dark picture that it is impossible to make a measurement.

In order to minimise as far as possible the impact of changes in the recording parameters on the measurement result, they should be adjusted so that the brightness level of the recorded image is as close as possible to the uncertainty *u*_reg_(*x*′) level obtained.

### 4.2. Uncertainty of Measuring the Object Image Position on Image Sensor, Resulting from Parameters of Camera and Optical System u_cam_(x′)

In order to define the measurement uncertainty results of camera parameters *u*_cam_(*x*′), it is necessary to establish the influence of sensor sensitivity level on the uncertainty value *u*_sns_(*x*′). The second factor will be the influence of lens focal length on the uncertainty of measuring the position of the object image on the image sensor with a constant scale of reproduction *u*_f_(*x*′).

#### 4.2.1. Uncertainty of Measuring the Object Image Position on Camera Sensor due to Actual Sensor Sensitivity *u**_sns_(x′)*

Each image sensor is characterised by basic sensitivity. Its increase is obtained by amplifying the electrical signals from the photosensitive cells of the sensor. The higher the set sensitivity, the higher the required gain value. As the sensitivity increases, the image noise level also increases. Since noise is a random factor, which causes slight blurring of the image contour sharpness, it should be expected that due to the increase of sensor sensitivity, the uncertainty of measuring the object image position will also be increased.

To verify this, a series of experimental measurements were performed for the gradually increasing sensor sensitivity. All the other acquisition parameters were adjusted in such a way that the brightness of the recorded image was always the same. The measurement results from laboratory tests are shown in Figure 5.

The sensor sensitivity increase is presented in relative units as a multiplication of basic sensitivity *h*_b_. Analogous to Figure 4, registrations made at different sensor sensitivity levels were marked with different colours.

The uncertainty *u*_sns_(*x*′) was determined as a standard deviation, analogously to Formula (24). The value of this uncertainty for both axes as a function of the sensor sensitivity level is shown in Figure 6.

Based on the characteristics presented in Figure 6, it can be observed that the uncertainty value *u*_sns_(*x*′) slightly increases, together with the increase of the image sensor sensitivity. For the analysed case, the increase in horizontal axis can be described by the following dependence (result in [μm]):(25)$$u_{\mathrm{sns}}\left(x^{\prime }\right)=\left(0.00489\pm 0.00072\right)\cdot \left(\frac{h}{h_{b}}\right)+\left(0.1349\pm 0.0069\right)$$
and analogously for the vertical axis:(26)$$u_{\mathrm{sns}}\left(y^{\prime }\right)=\left(0.0061\pm 0.0013\right)\cdot \left(\frac{h}{h_{b}}\right)+\left(0.125\pm 0.013\right).$$

Between the basic sensor sensitivity level and the highest possible level of this sensitivity, which can be set for the considered camera type, the measurement uncertainty level *u*_sns_(*x*′) increases by approximately 35%. It should be remembered that the presented results were obtained for a specific type of camera and lens. For another pair of devices, the results may be different. However, the experiment confirmed the supposition that the increase of measurement uncertainty *u*_sns_(*x*′) due to the increase of image sensor sensitivity should be expected.

#### 4.2.2. Uncertainty of Measuring the Object Image Position on Camera Sensor *u_f_(x′)*, the Source of which is Lens Focal Length

The analyses presented above were carried out for a constant scale of reproduction, using a lens with a relatively short focal length. However, vision measurements can be performed from different distances. In order to maintain a constant scale of reproduction, the lens focal length should be appropriately adjusted. It is obvious that such changes will affect the uncertainty of measuring the object image position on the camera sensor because the micro-vibrations of the substructure and air movements, which are the sources of stochastic deviation of the measurement result, will have a stronger influence on the measuring system when the distance between the measuring object and the camera is increased. Increasing the distance made it necessary to use a lens with a longer focal length. As the focal length increases, the view angle of the lens decreases. However, if the view angle of the lens is smaller, the occurring disturbances will have a stronger impact on the momentary displacements of the object image on the image sensor.

To determine the influence of these changes, the experimental measurements were performed for the focal lengths of the optical system and the stand parameters presented in Table 1.

The exposure parameters were selected so that the brightness of the registered image was the same for all lenses. The laboratory experiments were performed for medium settings of image sensor sensitivity. The obtained measurement results are presented in Figure 7 and Figure 8. Analogously to Figure 4 and Figure 5, in Figure 7, the registrations made for each focal length of the lenses are shown in different colours.

The measurement results show that the measurement uncertainty increases several times as the lens focal length increases with a constant scale of reproduction. In the considered case, this increase in the horizontal axis can be described by the following formula (result in [μm] for the focal length *f* given in [mm]):(27)$$u_{f}\left(x^{\prime }\right)=\left(0.000576\pm 0.000016\right)\cdot f+\left(0.1370\pm 0.0064\right)$$
and for the vertical axis respectively:(28)$$u_{f}\left(y^{\prime }\right)=\left(0.000782\pm 0.000037\right)\cdot f+\left(0.129\pm 0.015\right).$$

### 4.3. Combined Standard Uncertainty u(x′) and u(y′)

The combined standard uncertainty in measuring the object image position on the camera sensor in the horizontal axis *u*(*x*′) and vertical axis *u*(*y*′), described by Formula (23), as a function of changes in the acquisition image brightness and lens focal length, for the considered case, takes the values as shown in Figure 9.

The influence of changing the image sensor sensitivity is also marked in Figure 9, but it is minimal compared to the impact of the other factors. The solid line indicates the measured dependencies, and the dashed line indicates the estimated ones. While analysing the presented results, it can be concluded that the dominant factor affecting the level of uncertainty is the brightness of the recorded frame, which is the result of the registration parameters, i.e., shutter speed and lens aperture. To reduce the level of uncertainty, the brightness level of the registered image should also be set at the level as close as possible to the optimum one.

The second parameter, in terms of the impact on the measurement uncertainty level, is the lens focal length. However, it should be noted that this parameter results from the requirements related to the subject of measurement and the spatial configuration of the stand, so that most often, it is impossible to change it, or such a change is possible within a very limited range.

The change of the image sensor sensitivity has the smallest impact on the value of the combined standard uncertainty level. In comparison to other factors, it is essentially irrelevant, so that striving to reduce sensitivity at the expense of the brightness level of the registered image would be completely unjustified.

## 5. Influence of Optical System Parameters on Real Measurements Results, on the Example of Vertical Displacement of the Overhead Power Line

To illustrate how the chosen optical system parameters influence the real measurements results, the measurements of the vertical movements of a high-voltage overhead power line conductor near the centre of the suspension span have been performed, as shown in Figure 10.

The purpose of the measurement was only to show the influence of the selected parameter of the vision system on the measurement results. Therefore, the subject of measurement, i.e., the displacement of the high-voltage (HV) power line conductor, should be treated as an example. At the same time, the selected measurement object shows the possibilities of vision measurement systems, which allow for measuring the displacements of an object, which are practically unmeasurable in another way.

The measurements were performed from a relatively long distance (*k* ≈ 115 m), which required using a very long focus lens (*f* ≈ 800 mm). Assuming that typical laboratory measurement devices are used, the obtained level of combined standard uncertainty calculated in accordance with Dependence (5) can be estimated at *u*(*y*) ≈ 0.8 mm. The exact values of geometrical parameters are not important in this case, because the purpose of the measurement was to show the impact of changes in the registration parameters on the measurement results.

Measurements were performed for an optimal brightness level of the registered image and the situation when the brightness was 1.5 EV lower than in the optimal one. The optimal image brightness level was determined in the same way as in laboratory measurements. Measurement results are presented in Figure 11.

The experimental results (from Figure 10) presented in Figure 11 confirm the analysis that was presented in this paper. When measurements are performed for an image brightness level 1.5 EV lower than the optimal one, a significantly larger stochastic distribution of the obtained measurement results is visible, which is consistent with the information shown in Figure 3 and Figure 8. Consequently, a reduction of illumination level results in the uncertainty level increasing two or three times in relation to the optimal conditions.

## 6. Summary and Conclusions

The theoretical analysis presented in the article confirmed by a practical experiment carried out in laboratory conditions showed that the level of uncertainty in the position measurement performed with the use of an image camera significantly depends on numerous factors resulting from both the hardware configuration of the measurement system and from its parameters, e.g., camera settings.

Therefore, a fixed value of uncertainty cannot be given, as is the case with most conventional measuring instruments. 

As a result, it is justified to present in the article the methodology for determining the uncertainty of the object position on the image sensor.

It was also shown that the uncertainty analysis for video measurements is extremely important, as changes in camera and optical system parameters can change the level of achieved uncertainty even by one order of magnitude, which significantly affects the uncertainty of the final result for measurements carried out with vision techniques. Using these measurement methods, it is recommended to determine the expected level of uncertainty on the experimental path for a given case.

The conducted experiments showed that among the considered parameters, the level of image brightness has the greatest impact on the measurement uncertainty, as shown in Figure 9. Compared to it, the impact of sensor sensitivity changes is practically insignificant. In turn, the focal length of the lens most often depends on the specifics of the measured object and, as a rule, it cannot be chosen freely. Therefore, in order to ensure the lowest possible level of measurement uncertainty in a given case, it is first of all necessary to set the correct brightness of the recorded image by adjusting the aperture of the lens and the sensor sensitivity.

Determining the uncertainty in another way, e.g., by an analytical method, would be extremely troublesome, because it is impossible to designate individual components depending on Formula (23). The measurements will always be performed for a specific sensitivity level of the sensor, the focal length of the lens, and image recording parameters.

The analysis of the uncertainty of the position of the object in the vision systems presented in the article was performed with the assumption that the camera is not affected by any external factors such as changes in temperature, humidity, or additional vibrations. The analysis of the impact of these factors will be carried out as part of further experimental/research studies.

## Figures and Tables

**Figure 1 sensors-20-05433-f001:**
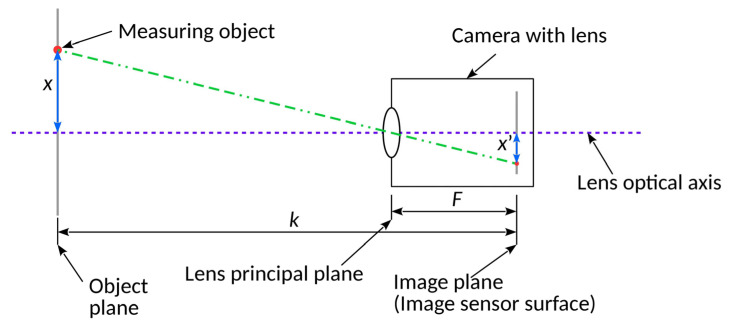
Scheme of the measurement stand (top view): *k*—distance between the object plane and the image plane (image sensor); *F*—distance between the plane of the optical centre of the lens and the image plane; *x*—displacement distance of the object measured in the horizontal axis in relation to the optical axis of the lens; *x*′—location of the image of the object measured in the horizontal axis.

**Figure 2 sensors-20-05433-f002:**
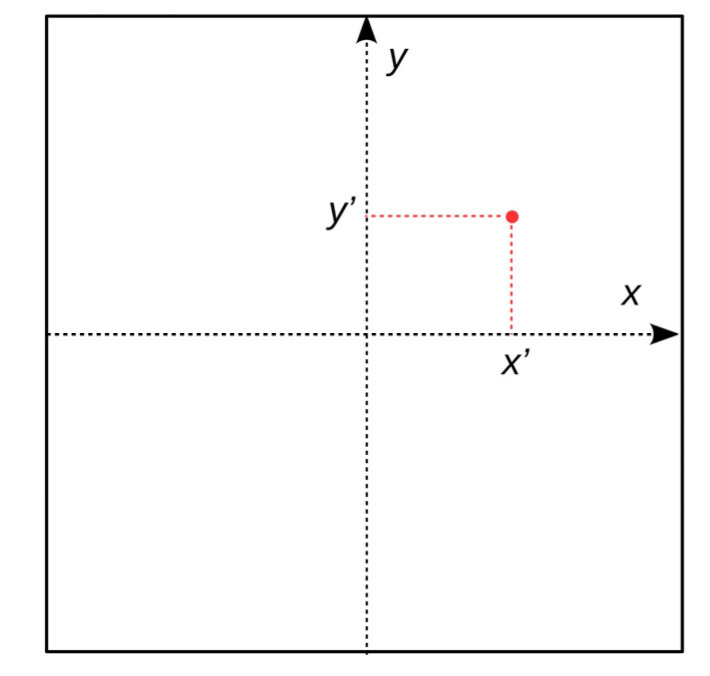
The subject of the analysis.

**Figure 3 sensors-20-05433-f003:**
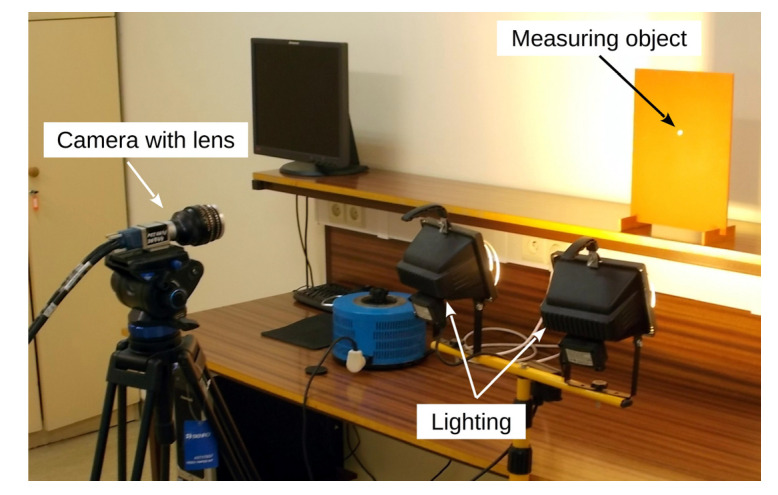
View of measuring stand.

**Figure 4 sensors-20-05433-f004:**
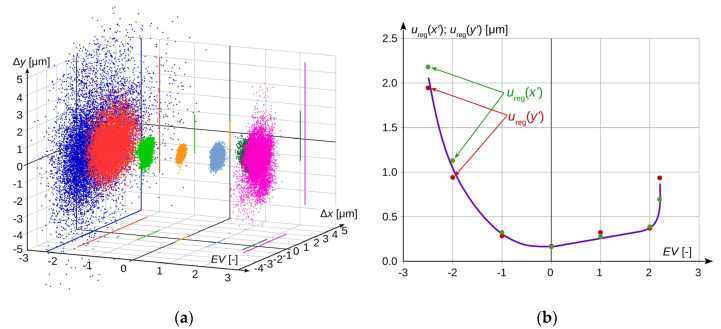
Influence of the image registration parameters: (**a**) stochastic distribution of the obtained measurement result depending on the brightness level of the recorded image; (**b**) uncertainty for the measurement in the horizontal axis *u*_reg_(*x*′) and vertical axis *u*_reg_(*y*′) in the function of changing the brightness of the recorded image.

**Figure 5 sensors-20-05433-f005:**
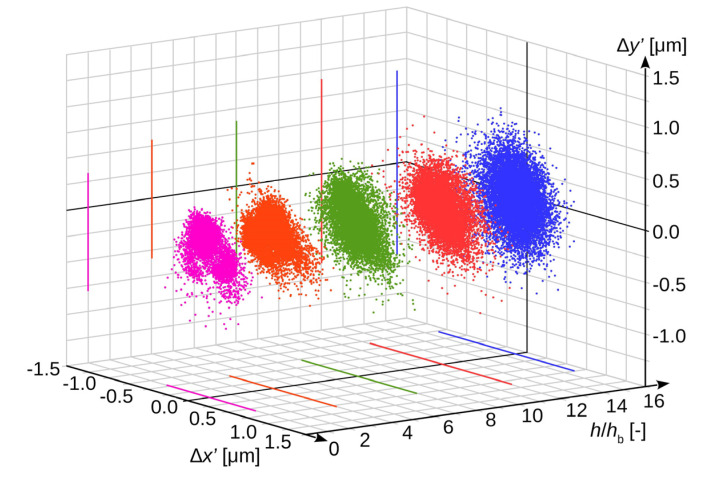
Stochastic distribution of the obtained results as a function of sensor sensitivity level.

**Figure 6 sensors-20-05433-f006:**
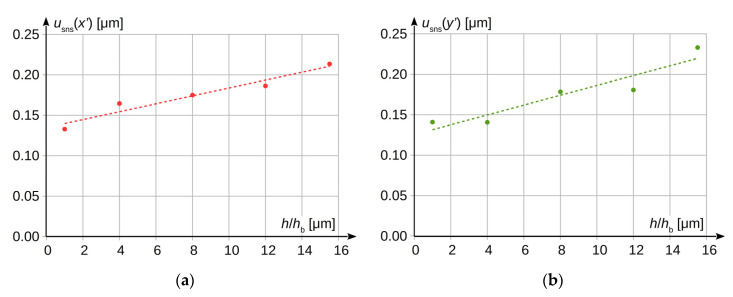
Dependence between uncertainty *u*_sns_ of measurement results from actual sensor sensitivity: (**a**) in horizontal axis *u*_sns_(*x*′); (**b**) in vertical axis *u*_sns_(*y*′).

**Figure 7 sensors-20-05433-f007:**
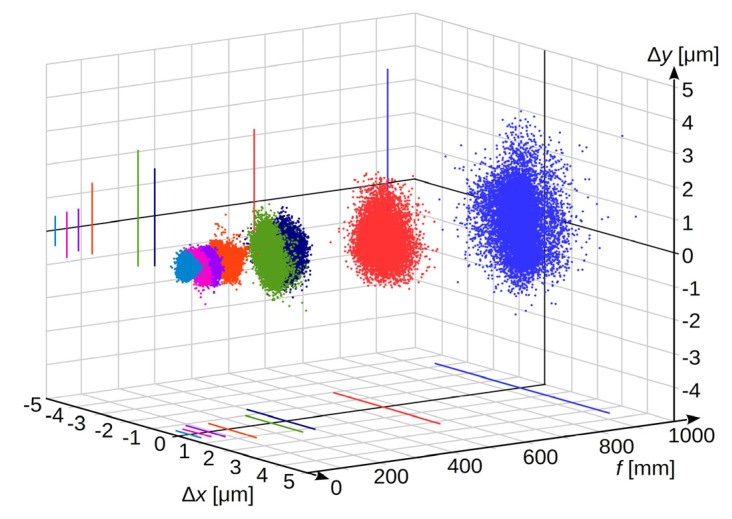
Stochastic deviation of measurement results as a function of lens focal length.

**Figure 8 sensors-20-05433-f008:**
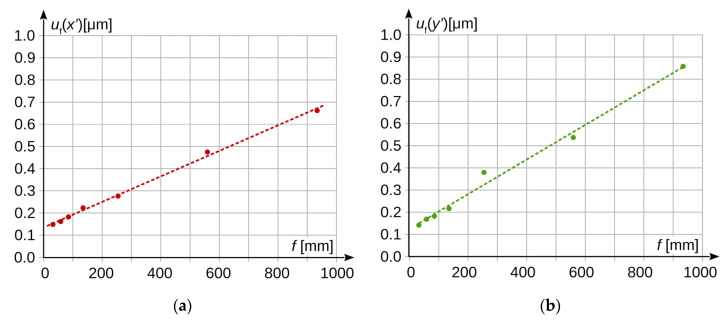
Standard measurement uncertainty as a function of lens focal length: (**a**) measurement results in the horizontal axis; (**b**) measurement results in the vertical axis.

**Figure 9 sensors-20-05433-f009:**
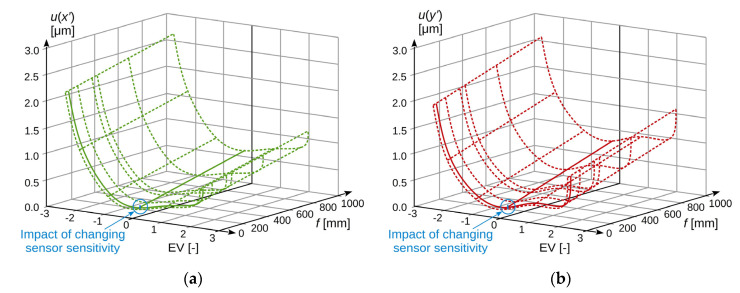
Standard uncertainty of measuring the position of the object image on the camera sensor as a function of changes in the acquisition image brightness and lens focal length: (**a**) measurement uncertainty in horizontal axis *x*; (**b**) measurement uncertainty in vertical axis *y*.

**Figure 10 sensors-20-05433-f010:**
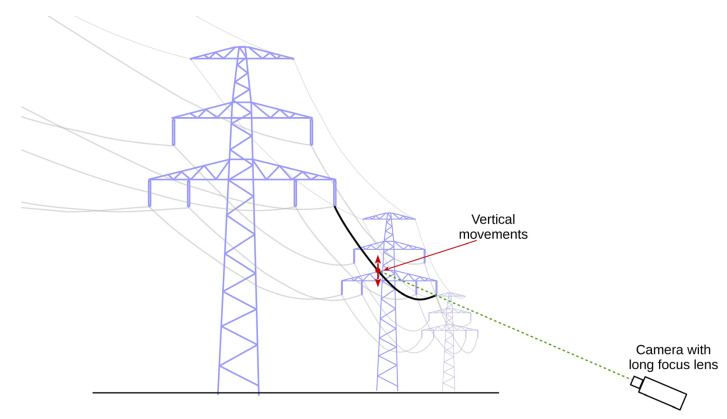
Measurement of the vertical movements of an HV overhead power line conductor—the principle of measurement.

**Figure 11 sensors-20-05433-f011:**
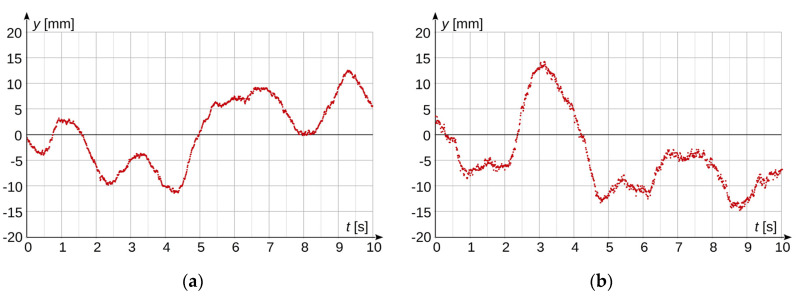
Measurement results of vertical movements of an HV overhead power line conductor; (**a**) for optimal brightness level; (**b**) for brightness level lower by 1.5 EV than the optimal.

**Table 1 sensors-20-05433-t001:** Technical data of measurement devices for checking the influence of lens focal length and measurement distance for uncertainty of measuring the position of the object image on the camera sensor.

No.	Lens Data (Optical Set)	Focal Length *f* [mm]		Distance *k* [mm]	Distance *F* [mm]
		Declared by Producer	Measured		
1	Lydith 3.5/30	30	32.044 ± 0.036	839.46 ± 0.87	33.370 ± 0.039
2	Helios 44-2 2/58	58	58.41 ± 0.04	1534.06 ± 0.87	60.827 ± 0.044
3	Jupiter 9 2/85	85	84.988 ± 0.046	2214.26 ± 0.87	88.53 ± 0.05
4	Jupiter 37 3.5/135	135	134.89 ± 0.06	3527.26 ± 0.87	140.485 ± 0.066
5	Telemegor 5.5/250	250	254.1 ± 0.1	6620.76 ± 0.87	264.70 ± 0.11
6	MC Sonnar 4/300	300	297.57 ± 0.12	7809.86 ± 0.87	309.86 ± 0.13
7	MC Sonnar 4/300 + K-6B 2x converter	600	559.12 ± 0.22	14,549.46 ± 0.87	582.43 ± 0.23
8	MC MTO-11 CA 10/1000	1000	933.84 ± 0.36	24,404.46 ± 0.87	972.61 ± 0.39

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
