# Peer review of "The Influence of Camera and Optical System Parameters on the Uncertainty of Object Location Measurement in Vision Systems"

_sensors, 2020, doi:10.3390/s20185433_

Round 1
Reviewer 1 Report
This paper presents an experimental study of the uncertainty associated with object measurement using a camera. The authors first present a method to identify the uncertainty of object location estimation on the imaging sensor by using two measured points. They then derive expressions for the uncertainty (standard deviation) of in-plane position measurements based on a decomposition of measurement error sources, and then evaluate the empirical accuracy of the uncertainty model for variations in illumination, gain and focal length. Finally, a short study of distance measurements of a suspended cable is provided to show degradation in measurement uncertainty with decreased brightness.
My first concern is that the English used to described the concepts is heavily flawed, with incorrect use of words such as analogical, throughout the paper, making understanding the work for a native English speaker quite difficult. Further, non-standard terms were used throughout the paper, camera matrix, focal length floating, adding to the confusion. The manuscript also had a large amount of yellow highlighting, often on words of questionable English quality. This was unclear and distracting.
Beyond this superficial concern, the work seems a clear evaluation of a well understood model of pixel measurement uncertainty. The method to determine this uncertainty via two measurements of a known object size seems clear, and is new to me. The main results though seem trivial, and the method is so specific to the camera and apparatus that the results are not easily transferrable to other devices, but must be reproduced for every new sensor and lens. The experimental results show the dependence of uncertainty on focal length (linear), sensor gain (linear), and image brightness. The first two seem obvious, as less light from increased focal length would increase SNR, as does signal amplification through gain. The third is interesting, and again matches intuition on SNR given under and over exposed images. It would be far more interesting if models for uncertainty were proposed that could predict these trends for a camera given a few measurements from a simpler apparatus.
Further, the detail on the experimental contributions to uncertainty was insufficient. It's not clear if the measurement error is caused by the print quality of the target or the sensor, or what the requirement is for this. It's not clear what the error analysis of the various measurements contributes to the final results for each of the three evaluated changes. There is no assessment of whether the uncertainties determined in experimentation match the model defined in the first half of the paper, no comparison of the distributions to Gaussians, no statistical tests to confirm the validity of measurements and models. It is very hard to draw conclusions from the work, other than the vague notions of increase/decrease w.r.t. the three parameters varied.
Author Response
This paper presents an experimental study of the uncertainty associated with object measurement using a camera. The authors first present a method to identify the uncertainty of object location estimation on the imaging sensor by using two measured points. They then derive expressions for the uncertainty (standard deviation) of in-plane position measurements based on a decomposition of measurement error sources, and then evaluate the empirical accuracy of the uncertainty model for variations in illumination, gain and focal length. Finally, a short study of distance measurements of a suspended cable is provided to show degradation in measurement uncertainty with decreased brightness.
My first concern is that the English used to described the concepts is heavily flawed, with incorrect use of words such as analogical, throughout the paper, making understanding the work for a native English speaker quite difficult. Further, non-standard terms were used throughout the paper, camera matrix, focal length floating, adding to the confusion. The manuscript also had a large amount of yellow highlighting, often on words of questionable English quality. This was unclear and distracting.
Authors Action/Response:
The authors agree with the Reviewer. The paper has been proofread. The yellow highlights indicate changes made in the previous version.
Beyond this superficial concern, the work seems a clear evaluation of a well understood model of pixel measurement uncertainty. The method to determine this uncertainty via two measurements of a known object size seems clear, and is new to me. The main results though seem trivial, and the method is so specific to the camera and apparatus that the results are not easily transferrable to other devices, but must be reproduced for every new sensor and lens. The experimental results show the dependence of uncertainty on focal length (linear), sensor gain (linear), and image brightness. The first two seem obvious, as less light from increased focal length would increase SNR, as does signal amplification through gain. The third is interesting, and again matches intuition on SNR given under and over exposed images. It would be far more interesting if models for uncertainty were proposed that could predict these trends for a camera given a few measurements from a simpler apparatus.
Authors Action/Response:
The authors thank the reviewer for a valuable comment. We agree with the reviewer that the issue of determining the uncertainty model for measuring the object position in vision systems and predicting trends in the tested parameters for the camera is important, necessary and interesting. However, the development of this type of model and its validation is a separate problem, the examination of which is planned in the next stage of our research. For this purpose, it will be necessary to conduct further observations and perform a series of experimental research.
Further, the detail on the experimental contributions to uncertainty was insufficient. It's not clear if the measurement error is caused by the print quality of the target or the sensor, or what the requirement is for this.
Authors Action/Response:
The authors thank the reviewer for this comment. In order to measure the position of the examined object's image on the image sensor, it must be contrasted with the background, so that the image processing algorithm would allow for determining its position on the image sensor. The information about this fact is included in the contents of the paper. To improve the readability of the article and dispel the doubts in this regard, the manuscript has been supplemented with a description of the individual stages of image processing.
It's not clear what the error analysis of the various measurements contributes to the final results for each of the three evaluated changes.
Authors Action/Response:
The submitted manuscript concerns the issue of estimating the uncertainty of object location measurement in vision systems, therefore it is not clear to the authors why the reviewer uses the term of “error”. The authors assume that the reviewer's comment is related to the estimation of measurement uncertainty.
The manuscript presents an analysis of the impact of three parameters: the level of image brightness, the camera sensor sensitivity and the focal length of the lens. First, the uncertainty of object positon measuring on the image sensor was estimated when each of the previously mentioned parameters was changed, and then (in chapter 4.3 of the manuscript) the combined uncertainty of the measurement has been determined.
The contribution of each of the uncertainties, analysed in the manuscript, to the combined uncertainty is visible in Fig. 9. The manuscript has been extended to comment on this.
There is no assessment of whether the uncertainties determined in experimentation match the model defined in the first half of the paper, no comparison of the distributions to Gaussians, no statistical tests to confirm the validity of measurements and models. It is very hard to draw conclusions from the work, other than the vague notions of increase/decrease w.r.t. the three parameters varied.
Authors Action/Response:
The authors thank the reviewer for this comment. In the next stage of the research, it is planned to develop a model for estimating the uncertainty of measuring the position of the object in vision systems and to predict trends in the parameters under study for the camera. Statistical tests will also be carried out to verify and validate the accuracy of the results obtained from the measurements and the developed model for estimating the measurement uncertainty.
Reviewer 2 Report
The authors present a methodology for quantification of object location identification uncertainty with a vision system, along with an associated experiment in the laboratory environment and use case overview (measurement of vertical movement of an overhead line conductor).
The experimental methodology is sound, but the presentation of the theory and results should be improved.
The concept of measurement uncertainty is central to the paper. However, it is not formally defined in the introduction (particularly in relation to image measurements).
The aspect of image processing is totally omitted, and it is not clear how exactly the measurements were done given images from the camera.
The measurements of Δx and Δy are central to the experiments, but they are not defined either in Nomenclature nor in the theoretical section.
The use of term “camera matrix” to mean “image sensor” is confusing, as camera matrix has a specific meaning in geometric calibration of cameras for vision applications.
Orientation of axes in Figure 2 is non-traditional, as one typically deals with x to the right and y down orientation.
The paper has to be proof-read, and language should be improved.
Author Response
The authors present a methodology for quantification of object location identification uncertainty with a vision system, along with an associated experiment in the laboratory environment and use case overview (measurement of vertical movement of an overhead line conductor).
The experimental methodology is sound, but the presentation of the theory and results should be improved.
The concept of measurement uncertainty is central to the paper. However, it is not formally defined in the introduction (particularly in relation to image measurements).
Authors Action/Response:
The authors thank the reviewer for this comment. The definition of the measurement uncertainty has been added to the contents of the paper. The whole article has been reviewed for the clarity. All changes are highlighted in yellow.
The aspect of image processing is totally omitted, and it is not clear how exactly the measurements were done given images from the camera.
Authors Action/Response:
The authors thank reviewer for a valuable comment. The article has been supplemented with a description of the recording and stages of image processing.
The measurements of Δx and Δy are central to the experiments, but they are not defined either in Nomenclature nor in the theoretical section.
Authors Action/Response:
The definitions of the terms Δx and Δy have been added to the nomenclature.
The use of term “camera matrix” to mean “image sensor” is confusing, as camera matrix has a specific meaning in geometric calibration of cameras for vision applications.
Authors Action/Response:
The authors thank the reviewer for a valuable comment. The reviewer is right. The term “camera matrix” has been used improperly. The terminology has been corrected throughout the article.
Orientation of axes in Figure 2 is non-traditional, as one typically deals with x to the right and y down orientation.
Authors Action/Response:
The authors are of the opinion that the classic orientation of the coordinate axes was used, which is characteristic for the first quarter of the system, i.e. x to the right and y up.
The paper has to be proof-read, and language should be improved
Authors Action/Response:
The authors agree with the reviewer. The paper has been proofreaded. All changes are marked in yellow.
Reviewer 3 Report
Evaluation of the measurement uncertainty is an import topic for machine vision. This manuscript presents the methodology for estimating combined standard uncertainty of measuring the object position with a vision camera, which is meaningful to analyze a machine vision system. However, the consideration of the system model and the discussion of the uncertainty are insufficient. The following issues should be clarified.
- For quantitative measurement with a camera, the influence of the lens distortion cannot be ignored [1, 2]. However, the manuscript considers only the linear camera model without lens distortion. The uncertainty from lens distortion should be discussed in the system model.
- Exposure value (EV) is a number that represents a combination of exposure time and lens aperture, which is discussed in Section 4.1. However, the first two paragraphs of Section 4.2.1 discuss the exposure time and lens aperture, too. This may make reader feel confuse. The dependence and independence of image brightness, EV, exposure time, lens aperture and sensor gain (also referred as matrix sensitivity in the manuscript) need to be further clarified at the beginning of Section 4.
- The manuscript should provide guidelines or concluding comments on how to select the parameters of the vision system when we want to achieve an accurate measurement.
- In a real experiment, e.g. the experiment in Section 5, how to determine the optimal brightness level of the image?
- In Section 5, only the influence of the EV is comparatively shown in Fig. 11. It is better to demonstrate the influence of other parameters, e.g. the work distance k and the focal length f.
- In Section 5, the vertical movements are measured. But why the vertical axis in Fig. 11 is labeled as delta_z.
Minor:
- The language should be polished to make the manuscript easier to read.
- In line 255, “matrix size: 11.26x112.26 mm” may be “11.26x11.26 mm”.
- In line 361, “Figure 5” may be “Figure 7”.
References
[1] Weng J, Cohen P, Herniou M. Camera calibration with distortion models and accuracy evaluation. IEEE Transactions on pattern analysis and machine intelligence 1992; 14(10): 965-980.
[2] Zhang Z. A flexible new technique for camera calibration. IEEE Transactions on pattern analysis and machine intelligence 2000; 22(11): 1330-1334.
Author Response
Evaluation of the measurement uncertainty is an import topic for machine vision. This manuscript presents the methodology for estimating combined standard uncertainty of measuring the object position with a vision camera, which is meaningful to analyze a machine vision system. However, the consideration of the system model and the discussion of the uncertainty are insufficient. The following issues should be clarified.
-
For quantitative measurement with a camera, the influence of the lens distortion cannot be ignored [1, 2]. However, the manuscript considers only the linear camera model without lens distortion. The uncertainty from lens distortion should be discussed in the system model.
Authors Action/Response:
The aim of the paper was to show the influence of selected system parameters on the uncertainty of object position measurement in vision systems. To eliminate the influence of optical distortions of the lens on the measurement results, lenses with a high degree of correction were used, so that the influence of their distortions on the measurement result was negligible and there was no need to take it into account. The information about the optical quality of the lenses has been included in the contents of the article.
-
Exposure value (EV) is a number that represents a combination of exposure time and lens aperture, which is discussed in Section 4.1. However, the first two paragraphs of Section 4.2.1 discuss the exposure time and lens aperture, too. This may make reader feel confuse. The dependence and independence of image brightness, EV, exposure time, lens aperture and sensor gain (also referred as matrix sensitivity in the manuscript) need to be further clarified at the beginning of Section 4.
Authors Action/Response:
The authors thank the reviewer for a valuable comment. In order to improve the readability of the paper, the description of the EV parameter dependence on the lens aperture value and the exposure time has been reogranised and extended.
-
The manuscript should provide guidelines or concluding comments on how to select the parameters of the vision system when we want to achieve an accurate measurement.
Authors Action/Response:
The authors thank the reviewer for this comment. The text of the article has been supplemented with suggestions which parameter has the greatest impact on the uncertainty of object location measurement in vision systems and how to select image recording parameters in order to obtain the best possible results under given conditions.
-
In a real experiment, e.g. the experiment in Section 5, how to determine the optimal brightness level of the image?
Authors Action/Response:
The authors thank the reviewer for this valuable comment. In real measurement, the optimal image brightness level was determined in the same way as in laboratory measurements. That is, the conditions for which the standard deviation of the stochastic distribution of the measurement result was the smallest were assumed as the optimal brightness level. The contents of the article have been supplemented with relevant information.
-
In Section 5, only the influence of the EV is comparatively shown in Fig. 11. It is better to demonstrate the influence of other parameters, e.g. the work distance k and the focal length f.
Authors Action/Response:
The authors agree with the reviewer that presenting the influence of all parameters would be better, but the specificity of the measured object (high-voltage overhead power line conductor) makes it impossible to test the impact of changing the distance k and the lens focal length f, for technical and safety reasons. However, the influence of these parameters on the measurement uncertainty was examined in laboratory conditions, so it should be assumed that it will be the same for real measurements, even more so, as the impact of changing the EV parameter for the measurements of the real object confirmed the previously obtained results of laboratory tests.
-
In Section 5, the vertical movements are measured. But why the vertical axis in Fig. 11 is labeled as delta_z.
Authors Action/Response:
The authors thank the reviewer for this comment. The symbol has been changed.
Minor:
-
The language should be polished to make the manuscript easier to read.
-
In line 255, “matrix size: 11.26x112.26 mm” may be “11.26x11.26 mm”.
-
In line 361, “Figure 5” may be “Figure 7”.
Authors Action/Response:
The authors agree with the reviewer. The editional errors and misspellings have been corrected.
References
[1] Weng J, Cohen P, Herniou M. Camera calibration with distortion models and accuracy evaluation. IEEE Transactions on pattern analysis and machine intelligence 1992; 14(10): 965-980.
[2] Zhang Z. A flexible new technique for camera calibration. IEEE Transactions on pattern analysis and machine intelligence 2000; 22(11): 1330-1334.
Authors Action/Response:
The authors thank the reviewer for his suggestions concerning the references. The bibliography list has been supplemented.
Round 2
Reviewer 1 Report
Thanks for the thoughtful responses and improvements to the paper, it is more clearly presented and the scope of the undertaking is more clear.
Reviewer 3 Report
All the comments have been responded. It is recommended to accept this paper.